# Nonuniform Heating Method for Hot Embossing of Polymers with Multiscale Microstructures

**DOI:** 10.3390/polym13030337

**Published:** 2021-01-21

**Authors:** Chih-Yuan Chang

**Affiliations:** Department of Mold and Die Engineering, National Kaohsiung University of Science and Technology, Kaohsiung 807, Taiwan; cychang@nkust.edu.tw; Fax: +886-7-381-4526 (ext. 15404)

**Keywords:** hot embossing, graphene-based heater, microstructures

## Abstract

The hot embossing of polymers is one of the most popular methods for replicating high-precision structures on thermoplastic polymer substrates at the micro-/nanoscale. However, the fabrication of hybrid multiscale microstructures by using the traditional isothermal hot embossing process is challenging. Therefore, in this study, we propose a novel nonuniform heating method for the hot embossing of polymers with multiscale microstructures. In this method, a thin graphene-based heater with a nonuniform heating function, a facility that integrates the graphene-based heater and gas-assisted hot embossing, and a roll of thermoplastic film are employed. Under appropriate process conditions, multiscale polymer microstructure patterns are fabricated through a single-step hot embossing process. The quality of the multiscale microstructure patterns replicated is uniform and high. The technique has great potential for the rapid and flexible fabrication of multiscale microstructure patterns on polymer substrates.

## 1. Introduction

In recent years, polymer micro- and nanostructure devices have attracted increasing interest owing to their many potential applications. One of the methods for fabricating large-area polymer micro- and nanostructure devices is the hot embossing process, where the structures of a master mold are copied onto a polymer substrate. This process is cost-effective and facilitates mass production; therefore, it is widely used for fabricating optical films, biochips, small sensors, etc. Thus far, many effective polymer hot embossing methods have been proposed and developed, such as plate-to-plate hot embossing [1,2,3], gas-assisted hot embossing [4,5,6,7], ultrasonic hot embossing [8,9,10,11], roller hot embossing [12,13,14,15], soft mold-based hot embossing [16,17,18,19], and rapid hot embossing [20,21,22]. Most of these methods are isothermal hot-embossing processes with a uniform heating system. The steps involved in these methods are heating, embossing, cooling, and demolding. To date, many studies have experimentally and numerically investigated the deformation and flow behavior of thermoplastic polymer during hot embossing [23,24,25,26]. The replicated pattern quality of the hot embossed part is significantly influenced by the process parameters, such as embossing pressure, embossing temperature, and holding time [27,28,29,30,31]. Furthermore, the effects of the geometric structures and size of the mold cavity on the hot embossing process are critical issues that need to be addressed, because the polymer replication ratio and filling height of the hot embossed product share a complicated nonlinear relationship with cavity size and density [32,33,34,35]. A nonuniform filling phenomenon is observed when a single embossing master mold contains multiple, nonuniform feature geometries. Nonuniform filling often leads to molding or demolding-related defects, such as residual stress, product warpage, damaged microstructures, and distorted features. Thus, effectively and uniformly replicating large-area multiscale microstructures is difficult in the polymer hot-embossing process. In addition, most traditional hot embossing machines have inherent problems because of the uniform heating system and pressing mechanism involved in using upper and lower hot plates or hot rollers. Consequently, multiple and nonuniform temperatures cannot set in traditional hot embossing machines for the hot embossing of large-area polymer microstructures with multiple scales or complex patterns. Furthermore, controlling and increasing the yield and stability of the polymer hot embossing process with multiscale microstructures are difficult.

To solve these problems, we present an effective, rapid, and nonuniform heating method for the hot embossing process of polymers with multiscale microstructures. To realize this method, a large-area thin stainless-steel mold with three different microstructure patterns of different scales was fabricated through standard nanoimprinting lithography, UV photolithography, and wet chemical etching process. A thin heater with three independent heating zones was prepared in advance using a screen-printing process with graphene-based conductive inks. A new gas-assisted hot-embossing facility was developed and reformed in-house [36]. The facility contained a traditional hot-embossing machine, a reel-to-reel mechanism, and a power supply unit with three independent outputs. During the polymer hot embossing process, the stack of a thin stainless steel mold and thin graphene-based heater was covered with polycarbonate (PC) film. The thin graphene-based heater with three heating zones was connected to the power supply unit and operated completely independently. Because the three heating zones of the thin heater work independently, different temperatures can be set depending on the various microstructure patterns of different scales. Thus, three microstructure patterns of different scales can be successfully and uniform replicated in a single-step hot embossing process.

In this study, the electric heating behavior of the thin graphene-based heater was tested and investigated. The uniform and nonuniform heating method of polymer hot embossing process with multiscale microstructures were conducted. The results of applying a uniform and nonuniform heating method for the hot embossing of polymers with multiscale microstructures were compared to verify the effectiveness of our proposed nonuniform heating method. Moreover, the surface morphology and replication quality of the fabricated microstructure patterns of different scales on the polymer substrates were analyzed.

## 2. Experimental Setup

### 2.1. Fabrication of a Master Mold with Multiscale Microstructure Patterns

In this experiment, a master mold with large-area and multiscale microstructure patterns was fabricated. Figure 1 shows the diagram and photograph of the master mold. The master mold was fabricated from stainless steel material (SUS 304L). The surface of the thin stainless-steel mold was divided into three regions, each with a different microstructure pattern. In each region, 25 small square areas formed the microstructure patterns. The microstructure patterns of region A, region B, and region C were a nanogroove array, microcircular hole array, and microgroove array, respectively. The width, pitch, and depth of nanogroove array were 500, 1000, and 500 nm, respectively. The diameter, pitch, and depth of the microcircular hole array were 100, 200, and 100 μm, respectively. The width, pitch, and depth of microgroove array were 100, 100 and 50 μm, respectively.

The fabricating procedure of the master mold with three different microstructure patterns is shown in Figure 2. First, a thin (0.15 mm thick) stainless steel plate (SUS 304L) was cleaned and chemically polished. Then, the surface of the stainless steel plate was divided into three regions; when one region was being processed, the other two regions were protected. Consequently, the microstructure patterns of region A, region B, and region C were sequentially manufactured. In the first stage, a nanogroove array pattern was fabricated on region A through standard nanoimprinting lithography and wet chemical etching. Here, a silicon mold with a nanopattern as a nanoimprinting tool was prepared using e-beam lithography and reactive-ion etching technology. After the nanoimprinting lithography process, the thin stainless-steel substrate with the imprinted-resist pattern was then soaked in an etchant during the etching process. The chemical composition of the etching solution was 10 wt% FeCl_3_, 15 wt% HCl, and 3 wt% HNO_3_. The wet etching process was conducted at 40 °C for 5 min, following which the nanogroove array pattern on the thin stainless-steel plate was obtained. In the second stage, a dry film photoresist (ETERTEC^®^ HT-115; Eternal Materials Co., Ltd., Taiwan) was laminated on region B of the stainless steel plate by using a dry-film laminator. The thin stainless-steel plate with the dry-film photoresist layer was then exposed to UV radiation through a photomask with a microcircular array pattern for 15 s using a UV-exposure machine. The photoresist layer was then dipped in a developer solution (1.2% Na_2_CO_3_·H_2_O) for 35 s, following which the microcircular hole array pattern on the photoresist layer was obtained. The thin stainless steel plate with the photoresist pattern was then soaked in an etchant (10 wt% FeCl_3_, 15 wt% HCl, and 3 wt% HNO_3_) during the chemical etching process. The wet etching process was conducted at 40 °C for 60 min. Then, the dry-film photoresist layer was stripped using a 5% NaOH solution. The microcircular hole array pattern on region B of the thin stainless steel plate was thus obtained. Finally, in the third stage, a microgroove array pattern was fabricated on region C of the thin stainless steel substrate. The fabrication process in the third stage was similar to that in the second stage; only the photomask pattern design and wet etching time were changed.

### 2.2. Fabrication of a Graphene-Based Heater with Nonuniform Heating Function

In recent times, graphene and graphene-based composites have attracted considerable attention for use in heater applications because of their unique properties, including high electrical conductivity, outstanding mechanical properties, large specific surface area, and high heat conductivity. In this study, a thin and large-area heater with nonuniform heating function was fabricated through a screen-printing process with graphene-based conductive inks. The screen printing inks were composed of graphene powders (45%) and epoxy resin (25%); these inks were prepared in-house. The viscosity of a graphene-based conductive ink is 18,000 ± 1000 cps. Figure 3 presents a design drawing and construction of the thin graphene-based heater with three heating zones. A polyimide film with high-temperature resistance was used as a printing substrate. After the screen-printing process, the graphene-based conductive ink was cured at 150 °C for 30 min. Then, copper sheets were glued to both ends of the graphene-based conductive layer, serving as electrodes for supplying the input voltage to the heater. To increase the service life of the thin graphene-based heater, a transparent epoxy resin was coated on the surface of the heater as a protective layer. In the experiment, the average thickness of graphene-based conductive layers was 20 ± 3 μm. The average electrical resistance of the graphene-based heaters was 35 ± 5 Ω sq^−1^, and their surface roughness was characterized through atomic force microscopy conducted at measurement points in the middle of the heaters. The average surface roughness and minimum surface roughness were 20.3 and 12.1 nm, respectively.

### 2.3. Gas-Assisted Hot Embossing Facility with a Nonuniform Heating System

Figure 4 presents the gas-assisted hot embossing machine for the hot embossing process of polymers with multiscale microstructures used in the experiments. The facility mainly consisted of an upper chamber, a lower embossing platform with circuit connection system and hydraulic cylinder unit, a high-pressure nitrogen gas system, a reel-to-reel mechanism of plastic film, and a power supply unit with three independent outputs. During the gas-assisted hot embossing process, the stack of master mold and thin graphene-based heater was fixed on the embossing platform. The thin graphene-based heater with three heating zones was connected with external electrical wires and a power supply unit. A part of the plastic film roll was controlled and placed on the stack of master mold and thin graphene-based heater by using tension rollers. After the chamber was closed, nitrogen gas was introduced into it to exert a low gas pressure (3 kg cm^−2^) over the plastic film to prevent the film from creasing. Simultaneously, the power supply unit was turned on and the graphene-based heater rapidly heated the stack of plastic film and master mold. When the desired processing temperature was reached, the gas pressure (30 kg cm^−2^) applied to the plastic film and master mold was increased for a period. The softened polymer material gradually filled the microstructure cavities on the master mold. At the end of the embossing procedure, the power supply unit was turned off and the temperature decreased to the initial state of room temperature. Finally, nitrogen gas was vented, the chamber was opened, and the film with microstructure patterns was removed from the master mold to allow a new processing cycle to be initiated. Thus, a roll of plastic film with microstructure devices was successfully fabricated. In addition, the total cycle time of the process was less than 3 min.

In the experiment, a roll of optical PC film was used as the hot embossing substrate. The glass transition temperature and thickness of the PC film were 140 °C and 0.18 mm, respectively. To further investigate the production quality of the hot embossing of polymers with multiscale microstructures, we applied different voltages to the graphene-based heater with three heating zones. The temperature distributions in each zone of the graphene-based heater were measured using a thermal imaging camera (DONHO & Co., Ltd., Taiwan). The practical embossing temperatures were determined using thermocouples placed in the embossing stack. The embossing temperatures at different temperature zones were controlled and adjusted in the hot embossing process. Finally, the shape and dimensions of the fabricated microstructure patterns were characterized using a scanning electron microscope (Hitachi S-3000N, Japan) and a laser scanning confocal microscope (VK-X1000; Keyence, Taiwan). Thus, a 2D surface profile and average cross-sectional area of the fabricated polymer microstructures were obtained. The average replication ratio of the fabricated microstructure was also estimated (replication ratio = cross-sectional area of fabricated microstructure/cross-sectional area of microstructure cavity of mold). In the experiments, all the cross-sectional area of fabricated polymer microstructures were measured at various locations of polymer substrates; each sample was measured at nine different measurement points.

## 3. Results and Discussion

### 3.1. Electrothermal Performance of the Graphene-Based Heater with Three Heating Zones

Three heating zones of the graphene-based heater were independently connected to the power supply unit to investigate the electrothermal performances of the graphene-based heater. The temperature quickly increased to a steady state when the graphene-based heater was operated at a certain input voltage. Figure 5 shows the temperature distribution of the graphene-based heater with the three heating zones during the experiments. A uniform temperature distribution was observed on the surface of each zone of the heater when an input voltage of 30 V was independently applied to zones A, B, and C. The average steady-state temperature of each zone of the heater was 180 ± 3 °C. These results prove the stability and uniformity of the graphene-based heater, which was deemed suitable for the polymer hot embossing process. Figure 6 shows the variation in the steady-state temperature of the graphene-based heater with applied voltage. The result shows that the steady-state temperature of the heater varied from 84 to 214 °C when the voltage was increased from 12 to 32 V. This result was quite consistent with the trend of electrical power formula, which is defined as P = V^2^/R, where P is the total electrical (heating) power of the graphene-based heater, R is the resistance of the heater, and V is the applied voltage. Therefore, the maximum steady-state temperature of the graphene-based heater increased dramatically as the applied voltage was increased. Furthermore, the heating rate of the graphene-based heater also increased as the voltage increased. The heating rate of the heater varied from 2.4 °C s^−1^ to 12.7 °C s^−1^ when the voltage was increased from 12 to 32 V. Based on the above study, the practical embossing temperature was controlled by adjusting the applied voltage.

Moreover, input voltages of 30, 25, and 20 V were applied to zones A, B, and C of the heater, respectively. As seen in Figure 7, the average steady-state temperatures of zones A, B, and C were 180 ± 3 °C, 145 ± 3 °C, and 116 ± 4 °C, respectively. The result indicates that the heater exhibits a nonuniform heating function and can be used in the hot embossing of polymers with multiscale microstructures.

### 3.2. Processing Conditions and Fabrication Quality of Hot Embossing Process of Polymers with Multiscale Microstructures

To study the process conditions of the hot embossing process of polymers and to verify the quality of the fabricated microstructure patterns of different scales, three process parameters—practical embossing temperature, embossing pressure, and embossing time—were investigated. In the preliminary experiment, the traditional isothermal polymer hot embossing process with uniform heating temperature was performed and investigated. In this case, the embossing pressure and time were maintained at 30 kg cm^−2^ and 120 s, respectively, and the applied voltage to the three zones of the heater was 28 V. After the hot embossing process, three regions with different types of microstructure patterns were fabricated on a polymer substrate, as shown in Figure 8. However, the average peak heights and replication ratios of the multiscale microstructure patterns were different. The mold cavity of region C with the microgroove array was completely filled, and the average replication ratio was 99.5%. The average replication ratio of the mold cavity of region B was 82.8%; a typical cylindrical hemisphere array pattern was formed on the polymer substrate. By contrast, the average replication ratio in the mold cavity of region A was only 68.8%; many surface defects were observed, and the mold was not sufficiently filled. These results demonstrate that the replication ratio of microstructure patterns with a large size was higher than that of microstructure patterns with a small size. This is possibly because of the smaller microstructure cavity, which renders the polymer filling process difficult and therefore makes a higher embossing temperature or pressure required to increase the replication ratio of the microstructure patterns on the polymer substrate. The processing conditions were an embossing pressure and time of 30 kg cm^−2^ and 120 s, respectively, and the input voltage to the three zones of the heater was increased from 28 to 30 V. Then, the practical embossing temperature of zones A, B, and C were found to be 179.5, 181.5, and 182.7 °C, respectively. The experimental results show that the replication ratios of the microstructure patterns of regions A, B, and C were 88.6%, 99.7%, and 99.8%, respectively. Region A, which contains the nanogroove array cavity, was still incompletely filled. Although region B and region C are completely filled, the polymer substrate with multiscale microstructures causes warpage and deformation. Demolding-related surface defects, such as damaged microstructures and distorted features, were observed on regions A, B and C, as shown in Figure 9. This experimental result indirectly proves that the nonuniform filling phenomenon causes residual stress and demolding defects in the polymer microstructure devices. Therefore, a nonuniform heating method for the polymer hot embossing process is required. Uniform filling of the polymer into multiscale microstructure patterns is achieved by setting different temperatures in the different regions of the stack of the master mold and polymer substrate. In the verification experiment, the embossing pressure and time were maintained at 30 kg cm^−2^ and 120 s, respectively, and the input voltages to zones A, B, and C of the heater were set to 31, 29, and 28 V, respectively. The practical embossing temperatures of zones A, B, and C were 185.5, 177.1, and 169.1 °C, respectively. As seen in Figure 10, the mold cavity of each region was completely filled, and the average replication ratio was higher than 99.5%. In addition, the polymer substrate was easily separated from the master mold, and no clear surface defects were observed on the substrate. The result demonstrated that uniform filling of the polymer and high replication quality of the multiscale microstructure patterns can be obtained by employing gas-assisted hot embossing with a nonuniform heating system.

## 4. Conclusions

In this study, a rapid and nonuniform heating method for hot embossing of polymers with multiscale microstructures was presented. A graphene-based heater with three independent heating zones was fabricated and tested; this heater facilitates rapid heating. In addition, a facility for integrating the graphene-based heater and gas-assisted hot embossing process was developed. The experimental results reveal that the electrothermal performance of the graphene-based heater was stable and uniform. The steady-state temperature of the heater and practical embossing temperature can be altered by varying the applied voltage under the following processing conditions: gas pressure = 30 kg cm^−2^, embossing time = 120 s, and the practical embossing temperature of zones A, B, and C were 185.5, 177.1, and 169.1 °C, respectively. Uniform filling of the polymer into multiscale microstructure patterns can be achieved. Large-area and multiscale microstructures patterns were successfully fabricated on the polymer substrate through a single-step hot embossing process. These results demonstrate that the nonuniform heating method for the polymer hot embossing process could be effective and flexible in the production of various microstructure pattern devices with high throughput.

## Figures and Tables

**Figure 1 polymers-13-00337-f001:**
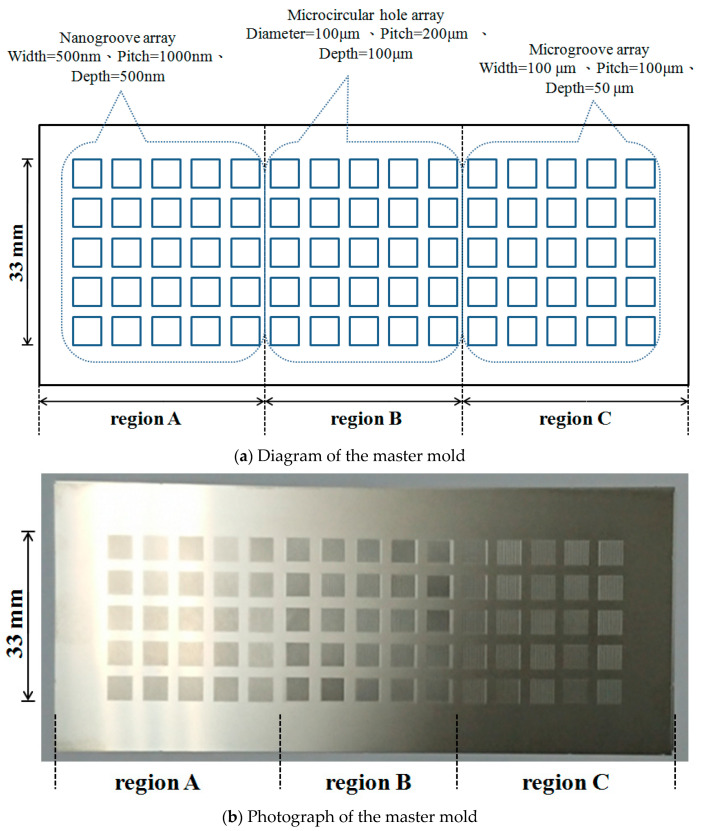
Diagram and photograph of the master mold with multiscale microstructures.

**Figure 2 polymers-13-00337-f002:**
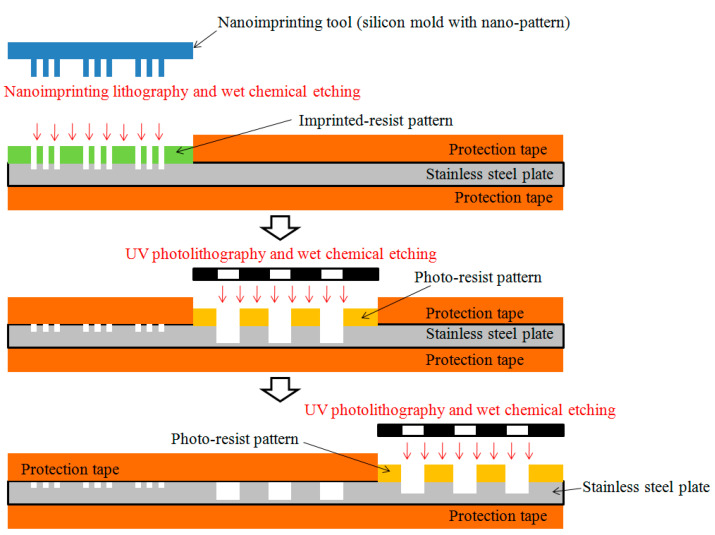
Procedure for fabricating the master mold with three different microstructure patterns.

**Figure 3 polymers-13-00337-f003:**
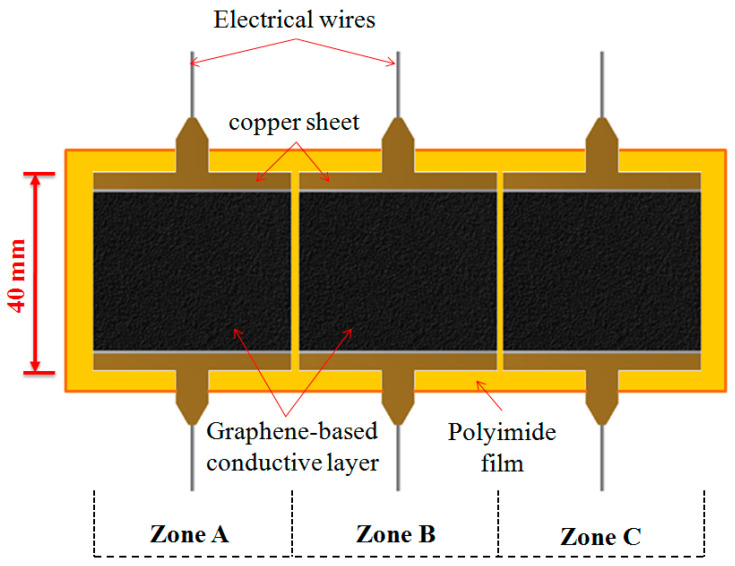
Design drawing and construction of the thin graphene-based heater with three heating zones.

**Figure 4 polymers-13-00337-f004:**
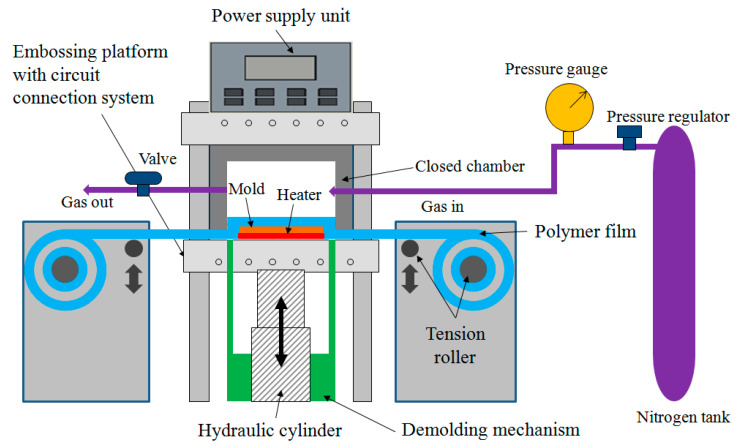
Drawing of the gas-assisted hot embossing facility with nonuniform heating system.

**Figure 5 polymers-13-00337-f005:**
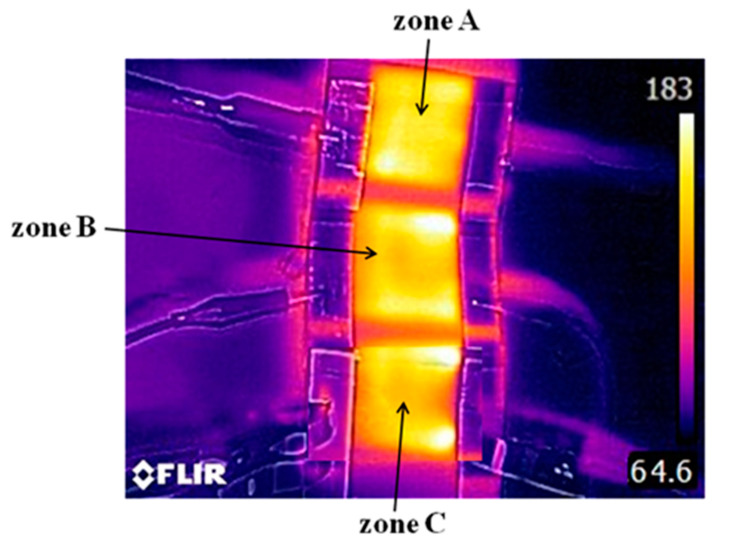
Temperature distribution of the graphene-based heater when an input voltage of 30 V was applied to each zone. (The average steady-state temperature of each zone of the heater was 180 ± 3 °C).

**Figure 6 polymers-13-00337-f006:**
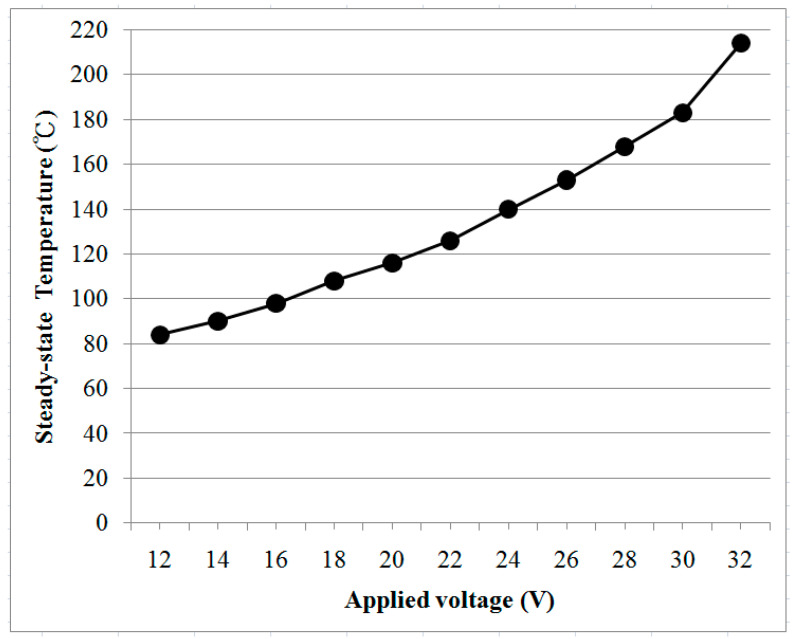
Steady-state temperature versus applied voltage of the graphene-based heater with three heating zones.

**Figure 7 polymers-13-00337-f007:**
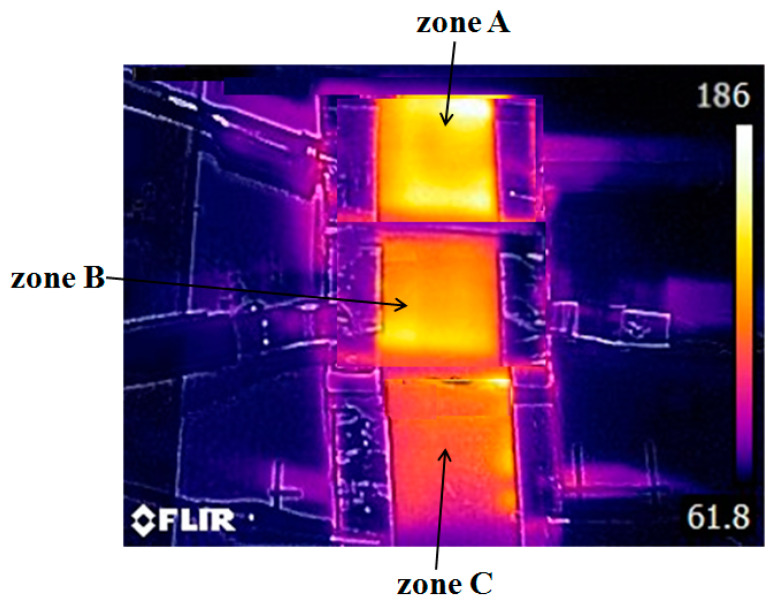
Temperature distribution of the graphene-based heater when input voltages of 30, 25, and 20 V were applied to zones A, B, and C, respectively. (The average steady-state temperatures of zones A, B, and C were 180 ± 3 °C, 145 ± 3 °C, and 116 ± 4 °C, respectively).

**Figure 8 polymers-13-00337-f008:**
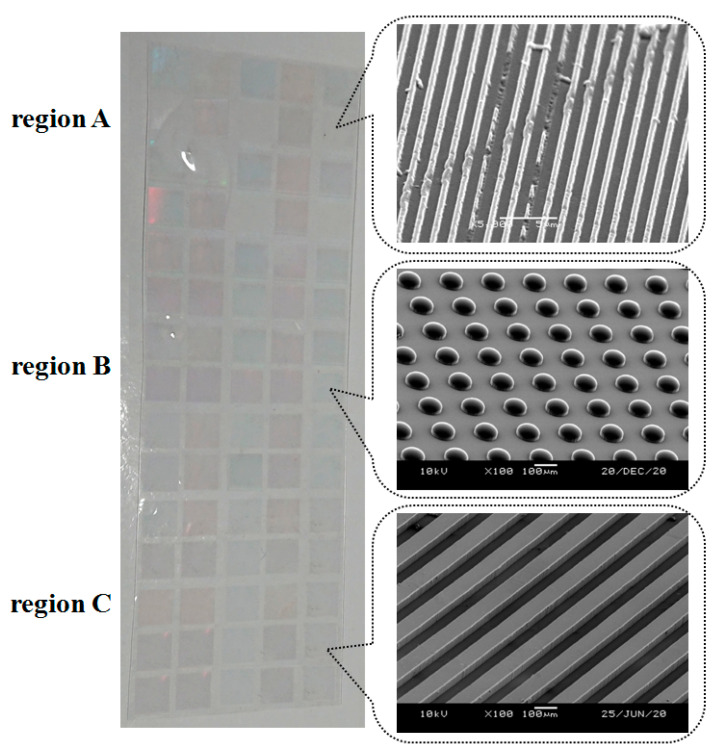
Preliminary experimental results of the traditional isothermal hot embossing process with multiscale microstructures. (When the embossing pressure and embossing time were maintained at 30 kg cm^−2^ and 120 s, the practical embossing temperatures of zones A, B, and C were 169.5, 169.2, and 168.8 °C, respectively.)

**Figure 9 polymers-13-00337-f009:**
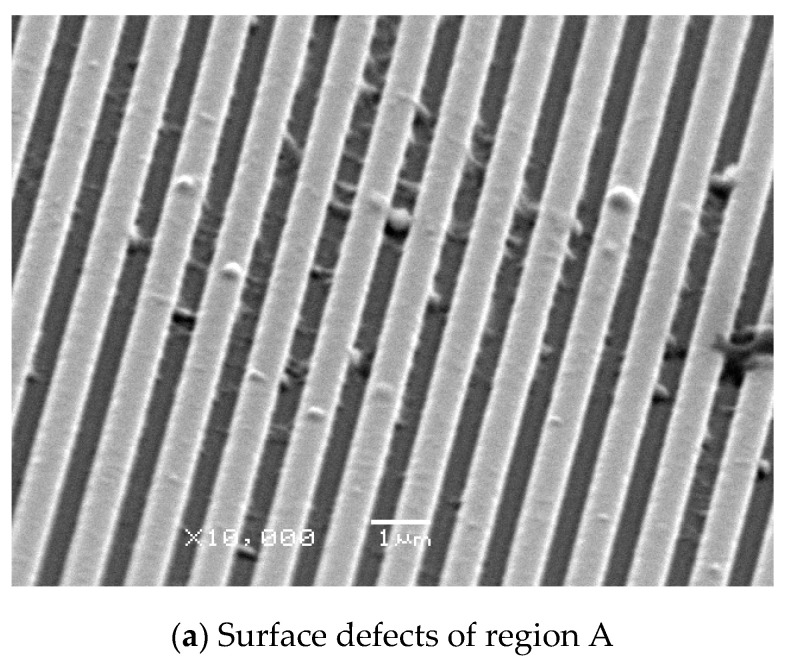
Surface defects in the microstructure patterns on region A, B, and C. (When the embossing pressure and embossing time were maintained at 30 kg cm^−2^ and 120 s, the practical embossing temperatures of zones A, B, and C were set to 179.5, 181.5, and 182.7 °C, respectively).

**Figure 10 polymers-13-00337-f010:**
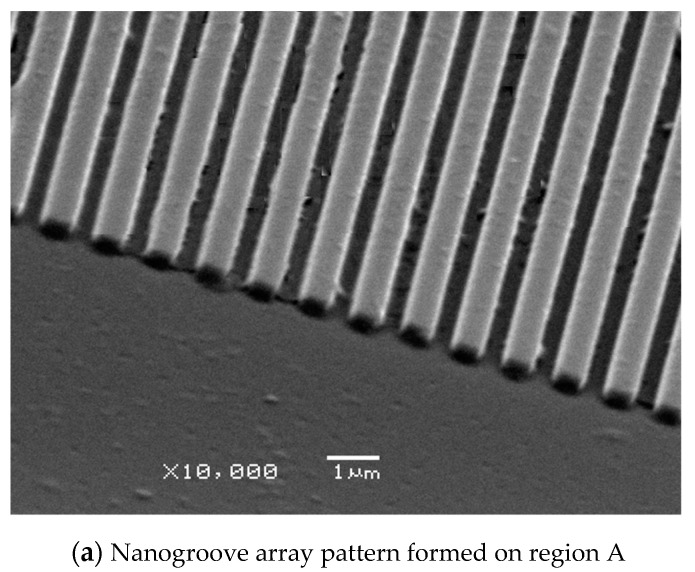
High replication quality of microstructure patterns on each region of a polymer substrate. (When the embossing pressure and embossing time were maintained at 30 kg cm^−2^ and 120 s, the practical embossing temperatures of zones A, B, and C were 185.5, 177.1, and 169.1 °C, respectively).

## Data Availability

Not applicable.

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
