# Peer review of "Nonuniform Heating Method for Hot Embossing of Polymers with Multiscale Microstructures"

_polymers, 2021, doi:10.3390/polym13030337_

Round 1

Reviewer 1 Report

In this article, the author proposed a novel nonuniform heating method for the hot embossing of polymers with multiscale microstructures. The paper has been written well. However, certain concerns need to be noticed and clarified: 1. The references are mostly old articles and the number of references is small. It is more necessary to add some latest references about the details of polymer hot embossing methods. 2. There are some grammatical mistakes in some parts of the article. Please check them carefully. For example, at line 154, "was" should be changed to "were". 3. In the Figure 9, the figure of the surface of region A should be shown to compare with the surface region B and region C. 4. In the conclusion part, the best technological conditions and parameters of the novel hot pressing method proposed by the author should be summarized.

Author Response

Dear reviewer,

Thank you very much for your suggestions that pointing out problems of the article.

This is very helpful to my research. I have tried my best to answer your questions and revise the paper.

Please find the attached file for your reference.

Reviewer 2 Report

The work of Chang entitled “Nonuniform heating method for hot embossing of polymers with multiscale microstructures” (Polymers-1084673) focuses on the topic of hot embossing of polymers using nonuniform heating method. It is an interesting topic, since the hot embossing are useful and effectively in replicating high-precision micro-scale structures of polymer substrate. The topic is well suitable of the journal of POLYMERS, and the article was well prepared and organized. This reviewer recommends its publication. However, there are some minor concerns are better to be re-considered by the author. For the benefit of the reader, it was suggested to address the following concerns.

  1. The “multiscale microstructures” is a confusing concept, and author use it in this article. Please explain and clarify it more clearly.
  2. In figure 6, there is only provide a line, and the temperature difference between the 12V and 32V is somewhat big. Please provide the detailed reason and compare them more.
  3. In the conclusion, author uses many words like “high-replication-quality”, “novel”, “effective”, “complex”, “high”, etc. These words should be concluded by the reader, not the author himself. More objective words are better to be used.

In conclusion, this reviewer suggests the author improves his article before its publication.

Author Response

Dear reviewer,

Thank you very much for your great care to the paper.

Your comments are very helpful to my research.

I have tried my best to answer your questions and revise the paper.

Please find the attached file for your reference.
